# Eliashberg Theory of a Multiband Non-Phononic Spin Glass Superconductor

**Giovanni Alberto Ummarino** [1,2] 

1   Istituto di Ingegneria e Fisica dei Materiali, Dipartimento di Scienza Applicata e Tecnologia,
    Politecnico di Torino, Corso Duca degli Abruzzi 24, 10129 Torino, Italy; giovanni.ummarino@polito.it
2   National Research Nuclear University MEPhI (Moscow Engineering Physics Institute), Kashira Hwy 31,
    115409 Moscow, Russia

**Abstract:** I solved the Eliashberg equations for a multiband non-phononic $s\pm$ wave spin-glass superconductor, calculating the temperature dependence of the gaps and of superfluid density. Their behaviors were revealed to be unusual: showing non-monotonic temperature dependence and reentrant superconductivity. By considering particular input parameters values that could describe the iron pnictide $EuFe_2(As_{1-x}P_x)_2$, a rich and complex phase diagram arises, with two different ranges of temperature in which superconductivity appears.

**Keywords:** superconductivity and magnetism; Fe-based superconductors; multiband Eliashberg theory; spin-glasses

## 1. Introduction

The discovery of the new iron-based superconductor family based on $EuFe_2As_2$ [1–8] allowed to investigate more deeply the interplay of magnetism and superconductivity. Compared to the past, there is now a new aspect to be considered: magnetism not only competes with superconductivity but also can be involved in the mechanism of superconductivity itself, as in the case of cuprates, heavy fermions and iron-based superconductors.

The case of the family of iron-based superconductors $EuFe_2As_2$ [1–8] is particularly interesting because the ferromagnetic and superconducting transition temperatures are close, where the first is connected to the ordering of $Eu2+$ local magnetic moments. It can also happen that the superconducting critical temperature is higher than that of magnetic ordering [1–8]. In these systems, a complex phenomenology of magnetic phases is observed: below the critical superconducting temperature, two distinct magnetic transitions take place; the ordering at higher temperature is associated with the antiferromagnetic interlayer coupling; whereas the behaviour at lower temperature might be identified as the change over to a spin-glass state, where the moments between the layers are decoupled [2,7]. Usually the spin-glass state [9] occurs in substitutionally disordered alloys [10–12], where, by means of the long-range Rudermann–Kittel–Kasuya–Yosida interaction, mediated by conduction electrons, the randomly distributed localized magnetic moments interact. Due to the fact that not all magnetic moments can be simultaneously satisfied in their spin orientation with respect to the others, it happens that frustration in the magnetic ordering arises. This fact produces an infinite number of random configurations that are degenerate in energy but separated by large energy barriers. In such a situation, the existing state cannot evolve into another (equally convenient from an energetic point of view) in experimentally accessible time scales. A freezing temperature $T_{SG}$ is associated with the spin-glass state, below which the spins freeze into one of these random configurations. The magnetic susceptibility in the spin-glasses shows a cusp at $T_{SG}$, while nothing happens to specific heat other than a broad maximum around $T_{SG}$ and no Bragg peaks, which usually

are a signal of long-range magnetic order, is found in neutron scattering experiments. The correct order parameter for these systems has to be related to the probability that a spin with a given direction at a finite time will have the same direction in the infinite-time limit. The frozen nature of the spin-glass state is reflected in this order parameter, but no spatial correlations are present as instead happens for other magnetic order parameters. Is it possible to reproduce this phenomenology connected with the superconducting state, coexisting with a spin-glass system, inside a theory? In this paper, I discuss the predictions of the theory on some physical quantities for a multiband spin-glass non-phononic $s\pm$-wave superconductor in the framework of Eliashberg equations, and I take as example the particular case of $EuFe_2(As_{1-x}P_x)_2$ [5]. Of course, I do not claim to fully reproduce the complex experimental phenomenology of this material but simply to obtain indications on the relevant input parameters that need to be included in the Eliashberg equations in the hope that one day it will be possible to find a material where only the superconducting state and the spin-glass state appear without any other complications. The starting point will be the theoretical work of M.J. Nass [13–15] and J.P Carbotte [16–19] that describe a single-band spin-glass s-wave phononic superconductor always in the framework of Eliashberg theory.

## 2. The Model

By introducing the order parameter for the spin-glass state as $q = \lim_{t \to +\infty} < \mathbf{S}_i(t) \cdot \mathbf{S}_i(0) >$, it is possible to describe mathematically this spin freezing [9]. This order parameter is proportional to the probability that a given spin that has a particular direction at $t = 0$ will still be orientated in that direction an infinite time later. This situation is quite different from having a order parameter in a ferromagnetic or antiferromagnetic system which reflects space as well as time correlations. Although each spin is essentially fixed in direction, in the absence of a magnetic field, upon averaging over all spins, the total spin is zero at all temperatures. By introducing a probability distribution, it is possible to reproduce the randomness of the exchange interaction and to then average over this distribution. It is necessary to use the replica approach in order to carry out averaging of the free energy over this distribution of exchange interactions and to succeed in finding a new order parameter defined as the configuration average of the equal time spin operators at a given site in different replicas of the system [9].

In the past papers [13–19], the developed theory is on phononic superconductors, where it was also added a contribution of antiferromagnetic spin fluctuations (dynamic part) and spin-glass (static part). In this case, it is not necessary to introduce the dynamic part (which is already considered since it is responsible for the pairing of electrons to form Cooper pairs) but only the static part, which is formally equal to the contribution of magnetic impurities with an additional dependence on temperature. The antiferromagnetic spin fluctuations have two components [13–19]: a dynamical component responsible for $s\pm$ superconductivity and a static component responsible for the spin-glass behaviour, that goes to zero at spin-glass critical temperature $T_{SG}$. For $T > T_{SG}$, the static component disappears and the material behaves like a normal $s\pm$ superconductor. In the old phononic low-temperature superconductors, the dynamic part is, usually, negligible and pair breaking, whereas in the multiband iron pnictide superconductors, it is the responsible for superconductivity. Contribution of the spin-glass phase can be represented in an approximate way in the Eliashberg equations by a term ($\Gamma^M(T)$) similar to that associated with the presence of magnetic impurities but with a temperature dependence. Precisely, the magnetic impurities scattering rate [16–19] that mimics the spin-glass state is $\Gamma^M(T) = \pi N(0) J^2 S^2 [1 - (\frac{T}{T_{SG}})^\beta]$, where $N(0)$ is the total density of states at the Fermi level, $J$ is a exchange constant, $S$ is the spin of the magnetic element and $\beta$ is a number [16–19] that can be 1 or 2 depending on the physical characteristic of the magnetic element ($Eu$ in this case) and on the host material (the specific iron pnictides). At this moment, there are not enough data to understand if $\beta$ is 1 or 2; therefore, I solve the Eliashberg equations in both cases. This theory stems from the desire to build a very simple model that still manages to grasp the fundamental physics of a multiband spin-glass superconductor. More sophisticated theories [20–24] start from multi-orbital Hubbard models that produce richer phase

diagrams and triplet superconductivity. For solving the Eliashberg equations, a lot of input parameters connected with the characteristic of the physical system are necessary. In the following, I will refer to $EuFe_2(As_{0.835}P_{0.165})_2$: a material belonging to the iron pnictide family. The electronic structure of the compound $EuFe_2(As_{0.835}P_{0.165})_2$ can be approximately described, in principle, as almost all electrons doped iron-based materials [25,26], by a three-band model with two electron bands (indicated in the following as bands 1 and 2) and one hole band (indicated in the following as band 3) [27]. In this way, the gap of the hole band, $\Delta_3$, has an opposite sign to the gaps residing on the electrons bands $\Delta_1$ and $\Delta_2$. For example, in the hole-doped iron compounds, within the five-orbital model [28], the Fermi surface comprises four sheets: two hole pockets around the point $(0,0)$ and two electron pockets around the points $(\pi,0)$ and $(0,\pi)$ [29], but usually, the two electron bands are very similar so it is possible to approximate the real situation with just a electron band with a density of states at the Fermi level that is the sum of the two contributions of the two bands. For the electron-doped materials, the opposite happens and it is possible to sum the contributions of the two hole bands. In both cases, in the end, it is possible to describe the superconductor with a 3-band model [25]. For completeness, it is necessary to mention an alternative approach to explain the superconductivity in the iron pnictides based on the appearance of the nontrivial Berry connection [30–32].

The compound $EuFe_2(As_{0.835}P_{0.165})_2$ is especially fascinating, since, despite the proximity of the magnetic and superconducting phases observed at rather high temperatures, there is just a little variation of their transition temperatures to these two phases [5]. The same happens for the stoichiometric material $RbEuFe_4As_4$, where the superconductivity and a long range magnetic orders exist independently from each other [33]. In this simple model, the effect of spin-glasses are simulated by some functions of temperature $\Gamma_{jk}^M(T)$ that go to zero before $T_c$ (precisely to $T_{SG} < T_c$), and in this way, they do not affect the critical temperature but change the behaviour of some physical quantities below $T_c$.

In the iron pnictides, the phonons are responsible for intraband coupling (*ph*) [26,34] and usually are neglected while the antiferromagnetic spin fluctuations (*sf*) are connected to interband coupling between holes and electrons bands ($s\pm$ wave model [26,34]). With the intention to reduce the number of free parameters, I use an effective two-band model (band 1 electrons and band 2 holes), where it is not possible to set to zero the intraband coupling and where the electron-boson coupling constants do not have an immediate interpretation [35,36] because this model simulates the true physical situation (three bands) with effective values of electron boson coupling constants in a two-band model. I investigate what happens in a multiband system, and for simplicity, I study a two bands system that simulates a real three-band system. In the following, the $s\pm$ wave two-band Eliashberg equations [37–40] are written. To calculate the critical temperature and the gaps, it is necessary to solve 4 coupled equations: 2 for the renormalization functions $Z_j(i\omega_n)$ and 2 for the gaps $\Delta_j(i\omega_n)$, where $j, k$ are band index (that range between 1 and 2) and $\omega_n$ are the Matsubara frequencies. The imaginary-axis equations [41–46] read as follows:

$$\omega_n Z_j(i\omega_n) = \omega_n + \pi T \sum_{m,k} \Lambda_{jk}^Z(i\omega_n, i\omega_m) N_k^Z(i\omega_m) + $$
$$+ \sum_k \left[ \Gamma_{jk}^N + \Gamma_{jk}^M(T) \right] N_k^Z(i\omega_n) \tag{1}$$

$$Z_j(i\omega_n)\Delta_j(i\omega_n) = \pi T \sum_{m,k} \left[ \Lambda_{jk}^\Delta(i\omega_n, i\omega_m) - \mu_{jk}^*(\omega_c) \right] \times$$
$$\times \Theta(\omega_c - |\omega_m|) N_k^\Delta(i\omega_m) + \sum_k [\Gamma_{jk}^N - \Gamma_{jk}^M(T)] N_k^\Delta(i\omega_n) \tag{2}$$

where $\Gamma_{jk}^N$ and $\Gamma_{jk}^M(T)$ are the scattering rates from nonmagnetic and magnetic impurities that, in this model, represent the term connected with the spin-glass phase. For spin-glass superconductors, the magnetic impurities scattering rates are $\Gamma_{jk}^M(T) = c_{jk}\pi N(0)J^2S^2[1 - (\frac{T}{T_{SG}})^\beta] = k_{jk}[1 - (\frac{T}{T_{SG}})^\beta]$, where $c_{jk}$ are weight connected with the bands ($\frac{c_{jk}}{c_{kj}} = \frac{N_k(0)}{N_j(0)}$ as the usual impurity scattering rates [41–43]) and, of course, $k_{jk} = c_{jk}\pi N(0)J^2S^2$. I set the nonmagnetic scattering rates $\Gamma_{jk}^N$ equal to zero because

I suppose good single crystals (no disorder). In the previous equations, I have $\Lambda_{jk}^Z(i\omega_n, i\omega_m) = \Lambda_{jk}^{ph}(i\omega_n, i\omega_m) + \Lambda_{jk}^{sf}(i\omega_n, i\omega_m)$ and $\Lambda_{jk}^{\Delta}(i\omega_n, i\omega_m) = \Lambda_{jk}^{ph}(i\omega_n, i\omega_m) - \Lambda_{jk}^{sf}(i\omega_n, i\omega_m)$, where

$$\Lambda_{jk}^{ph,sf}(i\omega_n, i\omega_m) = 2\int_0^{+\infty} d\Omega \Omega \alpha_{jk}^2 F_{jk}^{ph,sf}(\Omega)/[(\omega_n - \omega_m)^2 + \Omega^2],$$

$\Theta$ is the Heaviside function, and $\omega_c$ is a cutoff energy. The quantities $\mu_{jk}^*(\omega_c)$ are the elements of the $2 \times 2$ Coulomb pseudopotential matrix, and finally, $N_k^{\Delta}(i\omega_m) = \Delta_k(i\omega_m)/\sqrt{\omega_m^2 + \Delta_k^2(i\omega_m)}$ and $N_k^Z(i\omega_m) = \omega_m/\sqrt{\omega_m^2 + \Delta_k^2(i\omega_m)}$. The electron-boson coupling constants are defined as $\lambda_{jk}^{ph,sf} = 2\int_0^{+\infty} d\Omega \frac{\alpha_{jk}^2 F_{jk}^{ph,sf}(\Omega)}{\Omega}$.

In order to have the smallest number of free parameters and the simplest model that still grasps the physics of this system, I make further assumptions that have been shown to be valid for iron pnictides [41–43]. I assume, following Reference [34], that the total electron-phonon coupling constant is small (the upper limit of the phonon coupling in the usual iron-arsenide compounds is $\approx 0.35$ [47]), so I set, in first approximation, the phonon contribution equal to zero ($\lambda_{jk}^{ph} = 0$) and, following Mazin [26], the Coulomb pseudopotential matrix: $\mu_{jj}^*(\omega_c) = \mu_{jk}^*(\omega_c) = 0$ as well [26,41–43]. After all these approximations, I write the electron-boson coupling constant matrix $\lambda_{jk}$ in this way: [25,41–43]:

$$\lambda_{jk} = \begin{pmatrix} \lambda_{11}^{sf} & \lambda_{12}^{sf} \\ \lambda_{21}^{sf} = \lambda_{12}^{sf}\nu_{12} & \lambda_{22}^{sf} \end{pmatrix} \tag{3}$$

where $\nu_{12} = N_1(0)/N_2(0)$ and $N_j(0)$ is the normal density of states at the Fermi level for the $j$th band. Based on experimental data and theoretical calculations [41–43], I choose for the electron-antiferromagnetic spin fluctuation spectral functions $\alpha_{jk}^2 F_{jk}^{sf}(\Omega)$ a Lorentzian shape:

$$\alpha_{jk}^2 F_{jk}^{sf}(\Omega) = C_{jk}\left\{ \frac{1}{(\Omega + \Omega_{jk})^2 + Y_{jk}^2} - \frac{1}{(\Omega - \Omega_{jk})^2 + Y_{jk}^2} \right\}, \tag{4}$$

where $C_{jk}$ is a normalization constant, necessary to obtain the proper values of $\lambda_{jk}^{sf}$, while $\Omega_{jk}$ and $Y_{jk}$ are the peak energies and the half-widths of the Lorentzian functions, respectively [43]. Following the experimental data [48], I put $\Omega_{jk} = \Omega_0$, i.e., I assume that the characteristic energy of spin fluctuations is a single quantity for all the coupling channels, and $Y_{jk} = \Omega_0/2$. The spectral function used here, normalized to one, is shown in inset (a) of Figure 1.

The factors $\nu_{jk}$ in the definition of $\lambda_{jk}$ (Equation (3)) are unknown so I assume that they are equal, for example, to the $Ba(Fe_{1-x}Rh_x)_2As_2$ electron-doped case [49] so $\nu_{12} = 0.8333$ as well as the coupling constants [49], and I change slightly just the value of $\lambda_{22}$ for obtaining the correct critical temperature $T_c = 22$ K. At the end, the values are $\lambda_{11} = 1.00$, $\lambda_{12} = -0.17$ and $\lambda_{22} = 2.65$ for an averaged coupling constant $\lambda_t = \frac{\Sigma_{jk} N_j(0)\lambda_{jk}}{\Sigma_j N_j(0)} = 1.75$. For iron pnictides, it was experimentally found [50,51] that the empirical law $\Omega_0 = 2T_c/5$ holds; therefore, the value of the energy peak $\Omega_0$ of the Eliashberg spectral functions $\alpha_{jk}^2 F_{jk}^{sf}(\Omega)$ is fixed. To finish, in the numerical calculations, I used a cutoff energy $\omega_c = 180$ meV. These input parameters produce, by numerically solving the Eliashberg equations, exactly a critical temperature of 22 K.

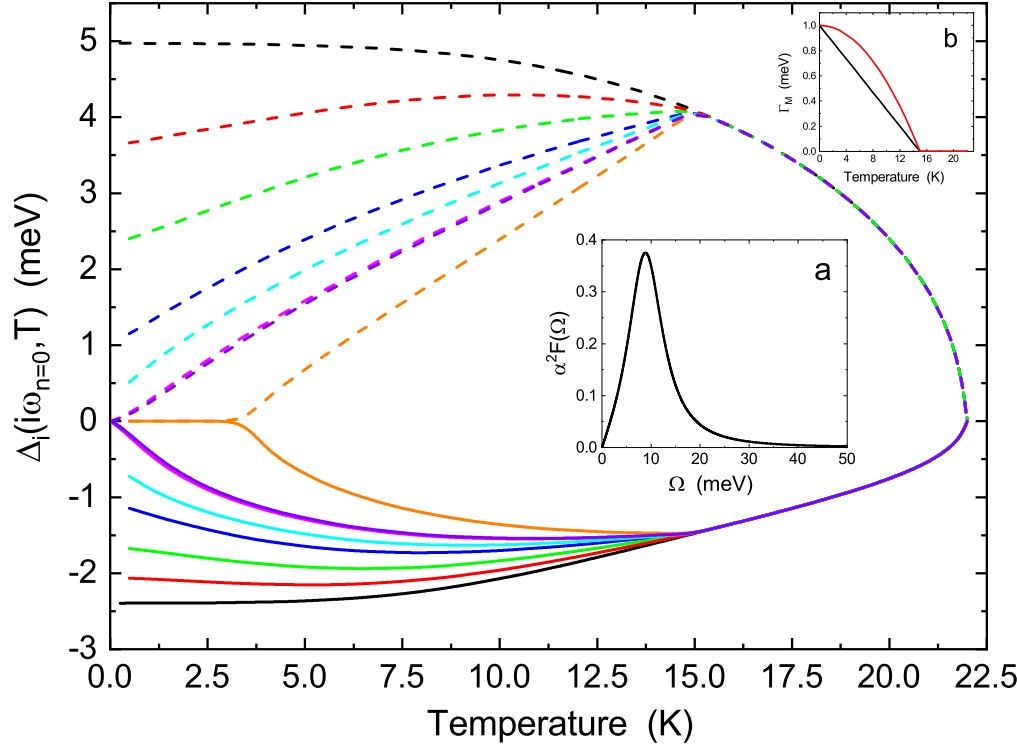

**Figure 1.** (Color online) The gaps $\Delta_i(i\omega_{n=0})$ in function of temperature obtained by solving the Eliashberg equations on an imaginary axis: solid lines are for $\Delta_1(i\omega_{n=0})$, and dashed lines are for $\Delta_2(i\omega_{n=0})$ in the case $k_{11} = k_{12} = 0.2k_{22}$ and $\beta = 1$. Black lines are for $k_{22} = 0$ meV, red lines are for $k_{22} = 1$ meV, green lines are for $k_{22} = 2$ meV, dark blue lines are for $k_{22} = 3$ meV, cyan lines are for $k_{22} = 3.5$ meV, magenta lines are for $k_{22} = 4$ meV, violet lines are for $k_{22} = 4.05$ meV and orange lines are for $k_{22} = 5$ meV. In inset (**a**), the antiferromagnetic spin fluctuation function, normalized to one, is shown, while in inset (**b**), the temperature dependence of $k_{jk}$ is shown (the black solid line represents $\beta = 1$, and the red solid line represents $\beta = 2$) with $k_{jk} = 1$ .

## 3. Calculation of Superconducting Gaps

In the iron pnictides, usually, the impurities are almost all concentrated in one band, i.e., in the hole band for the electron-doped materials as this case and in the electron band [52–54] for the hole-doped materials [55]. This means that, in the electrons-doped materials, $k_{22} >> k_{11}, k_{12}$. It is possible also to include orbital degrees of freedom that lead, in the BCS formalism, to a fully gapped $s \pm wave$ state very fragile [56] against impurities, while the experiments revealed that the suppression of $T_c$ is weaker than expected [57,58] for this model. The agreement with the experiment is found by considering strong coupling effects [59] and by going beyond the Born approximation when, in the materials, a large amount of impurities is present [49,60,61]. Indeed, the pure interband Eliashberg theory, also in the limit of weak coupling, is different from BCS theory [62]. I choose $k_{11} = k_{12} = 0.2k_{22}$ as happen in the $Ba(Fe_{1-x}Co_x)_2As_2$ [52]. By using the typical parameters of iron pnictides and spin-glass systems, I find that $k_{22} \simeq 3.1$ meV ($N(0) = 5.6$ states/eV, $S = 7/2$, $J = 0.12$ meV and $T_{SG} = 15$ K) [5,63]. Because the true values of the parameters in the last bracket are just approximate, I solve the Eliasberg equations for values close to 3.1 as $k_{22} = 0, 1, 2, 3, 3.5, 4, 4.05, 5$ meV in the two cases: $\beta = 1$ and $\beta = 2$. In the ideal case, it would be necessary to know the law that links $T_{SG}$ at the value of $k_{22}$. Here, $T_{SG} = 15$ K is an experimental input [5]. In Figures 1 and 2, the temperature dependence of gaps $\Delta_{1,2}(i\omega_{n=0})$ are shown. It is possible to see in Figure 1 that the absolute values of the gaps with increasing temperature at first increases until $T_{SG}$ and then decreases. This behaviour appears when magnetic impurities

(also without temperature dependence) and disorder are present, both in $s + +$ and $s\pm$ superconductor or if only disorder is present in $s\pm$ two bands superconductor [64,65].

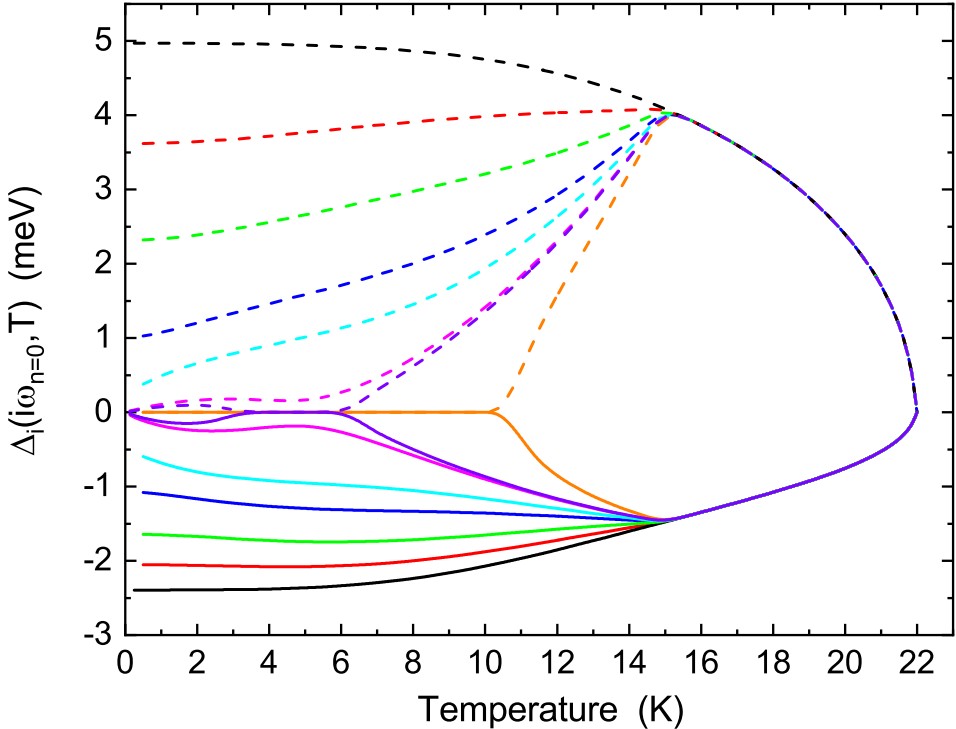

**Figure 2.** (Color online) The gaps $\Delta_i(i\omega_{n=0})$ in function of temperature obtained by solving the Eliashberg equations on imaginary axis: solid lines are for $\Delta_1(i\omega_{n=0})$, and dashed lines are for $\Delta_2(i\omega_{n=0})$ in the case $k_{11} = k_{12} = 0.2k_{22}$ and $\beta = 2$. Black lines are for $k_{22} = 0$ meV, red lines are for $k_{22} = 1$ meV, green lines are for $k_{22} = 2$ meV, dark blue lines are for $k_{22} = 3$ meV, cyan line are for $k_{22} = 3.5$ meV, magenta lines are for $k_{22} = 4$ meV, violet lines are for $k_{22} = 4.05$ meV and orange lines are for $k_{22} = 5$ meV.

For $k_{22} = 5$ meV and $\beta = 1$, reentrant superconductivity is obtained. As it is shown in Figure 2, in the case of $\beta = 1$, the effect is similar but stronger and, besides having reentrant superconductivity for $k_{22} = 5$ meV, it is possible to see an even more complex situation for $k_{22} = 4.05$ meV. In the last case, the system has three different critical temperatures: I think that it would be difficult to observe this behaviour in a real system because it arises from fine tuning of the input parameter. In an s-wave superconductor, the magnetic order destroys the superconductivity, but the increasing temperature weakens both the magnetic order and the coupling between the electrons in the Cooper pairs so the "reentrant" behaviour can emerge from the balance between the effect of magnetism and temperature. The reentrant superconductivity appears also in the single band case [17–19], but for a more realistic and complete model such as the one proposed in this paper, the phase diagram is richer and more complex. I solved the Eliashberg equations, for completeness, also in the case $k_{12} = 0.2k_{22}$ and $k_{11} = k_{22}$ always with $\beta = 1$ and $\beta = 2$. The results are shown in Figure 3 and are similar to previous ones in the general trend as a function of the value of $k_{22}$. In all cases, of course, for $T > T_{SG}$, the effect of "magnetic impurities" disappeared and the behaviour is the same as a standard two-band superconductor.

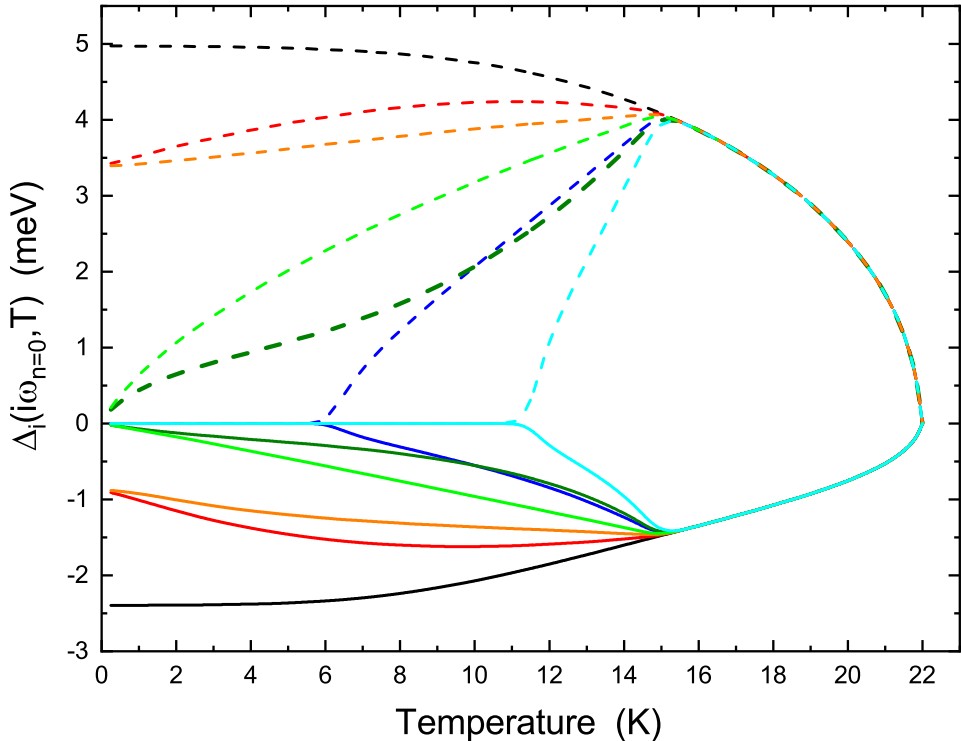

**Figure 3.** (Color online) The gaps $\Delta_i(i\omega_{n=0})$ in function of temperature obtained by solving the Eliashberg equations on imaginary axis: solid lines are for $\Delta_1(i\omega_{n=0})$, and dashed lines are for $\Delta_2(i\omega_{n=0})$ in the case $k_{11} = k_{22} = 5k_{12}$. Black lines are for $k_{22} = 0$ meV, red lines are for $k_{22} = 1$ meV and $\beta = 1$, orange lines are for $k_{22} = 1$ meV and $\beta = 2$, green lines are for $k_{22} = 3$ meV and $\beta = 1$, olive lines are for $k_{22} = 3$ meV and $\beta = 2$, dark blue lines are for $k_{22} = 5$ meV and $\beta = 1$, and magneta lines are for $k_{22} = 5$ meV and $\beta = 2$.

## 4. Calculation of the Penetration Depth

The penetration depth (or the superfluid density as it is shown in Figures 4–6) can be computed starting from the renormalization functions $Z_j(i\omega_n)$ and the gaps $\Delta_j(i\omega_n)$ by using the following formula [66]:

$$\lambda^{-2}(T) = \left(\frac{\omega_p}{c}\right)^2 \sum_{j=1}^{2} w_j \pi T \sum_{n=-\infty}^{+\infty} \frac{\Delta_j^2(\omega_n) Z_j^2(\omega_n)}{[\omega_n^2 Z_j^2(\omega_n) + \Delta_j^2(\omega_n) Z_j^2(\omega_n)]^{3/2}} \tag{5}$$

where $\omega_{p,i}$ is the plasma frequency of the *i*th band and $\omega_p$ is the total plasma frequency in order that $w_j = \left(\omega_{p,j}/\omega_p\right)^2$ is the weights of the single bands.

The low-temperature value of the penetration depth $\lambda_L(0)$ should, in principle, be related to the plasma frequency by $\omega_p = c/\lambda_L(0)$ [67] and appears as a multiplicative factor of the summation. Here, $w_1 = 0.72$ and $w_2 = 0.28$ as in the Co-doped iron compounds [52]. In principle, the calculation of superfluid density (penetration depth) is important in order to compare theoretical predictions with the experiment because it is easier to find these measurements in the literature [61]. In Figures 4 and 5, the superfluid density in function of temperature is shown when $k_{11} = k_{12} = 0.2k_{22}$, $k_{22} = 0, 1, 2, 3, 3.5, 4, 4.05, 5$ meV with $\beta = 1$ and $\beta = 2$. In Figure 6, I show the superfluid density when $k_{12} = 0.2k_{11} = 0.2k_{22}$, $k_{22} = 0, 1, 3, 5$ meV with $\beta = 1$ and $\beta = 2$. These results are a clear prediction of possible situations that can be easily identified. Unfortunately, there is still no experimental data to compare with these theoretical predictions. The behavior of the penetration depth as a function of temperature shows how the presence of a spin-glass state in competition with superconductivity substantially changes the phase diagram of a superconductor, making it extremely richer. Also, for

superfluid density, it is clear when reentrant superconductivity appears as well as the case of three critical temperatures (see Figure 5 violet line). In Figures 1 and 2, in the cases with $k_{22} = 4.00$ meV and $k_{22} = 4.05$ meV, the values of $\Delta_1(i\omega_{n=0})$ and $\Delta_2(i\omega_{n=0})$ at very low temperatures are almost zero but the corresponding superfluid density at the same temperatures is different from zero in an appreciable way. How is this possible? From the numerical solution of Eliashberg equation in the standard case (when the $k_{jk}$ are equal to zero), the maximum value of $|\Delta_i(i\omega_n)|$ is for $n = 0$ and $|\Delta_i(i\omega_n)|$ decreases when $|n|$ increases while, when $k_{jk}$ is different from zero, the dependence from $|n|$ is different and not usual. In the inset of Figure 5, the calculated values of $\Delta_1(i\omega_n)$ and $\Delta_2(i\omega_n)$ in the $k_{22} = 4.05$ meV and $\beta = 2$ cases at $T = 0.125$ K in function of $n$ is shown. In this case, it is possible to see that the dependence of $\Delta_j(i\omega_n)$ from $n$ is not standard. In this case, the maximum value of $|\Delta_i(i\omega_n)|$ is found for $n = 0$, so also if $\Delta_i(i\omega_{n=0}) \simeq 0$ meV, the corresponding superfluid density can be different from zero.

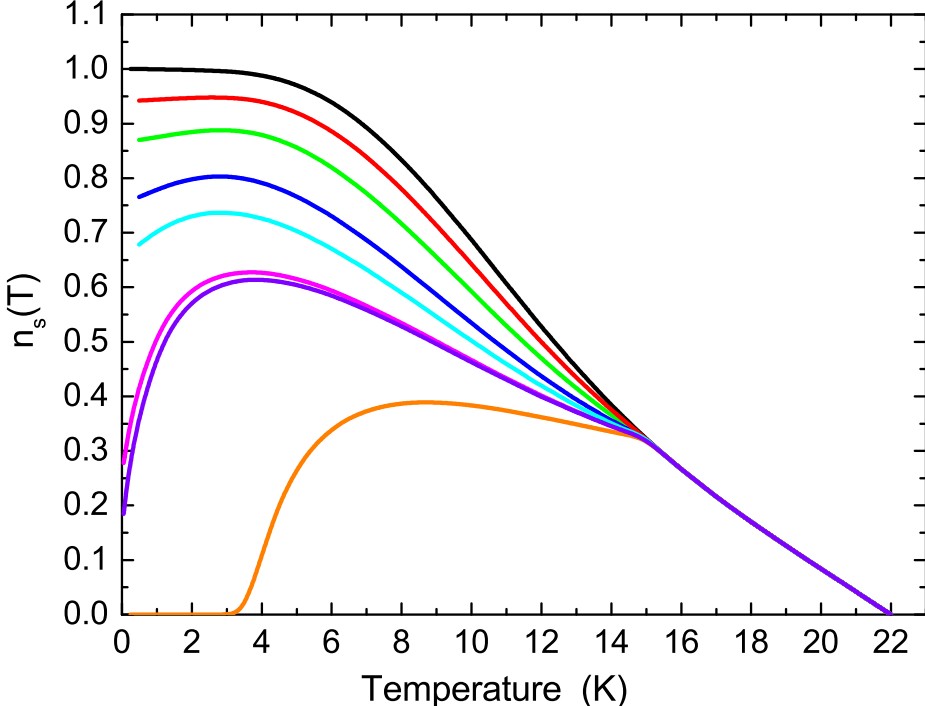

**Figure 4.** (Color online) The superfluid density $n_s(T)$ normalized at the value at T = 0 K in the case $k_{22} = 0$ in function of temperature obtained by solving the Eliashberg equations on an imaginary axis in the case $k_{11} = k_{12} = 0.2k_{22}$ and $\beta = 1$: black lines are for $k_{22} = 0$ meV, red lines are for $k_{22} = 1$ meV, green lines are for $k_{22} = 2$ meV, dark blue lines are for $k_{22} = 3$ meV, cyan lines are for $k_{22} = 3.5$ meV, magenta lines are for $k_{22} = 4$ meV, violet lines are for $k_{22} = 4.05$ meV and orange lines are for $k_{22} = 5$ meV.

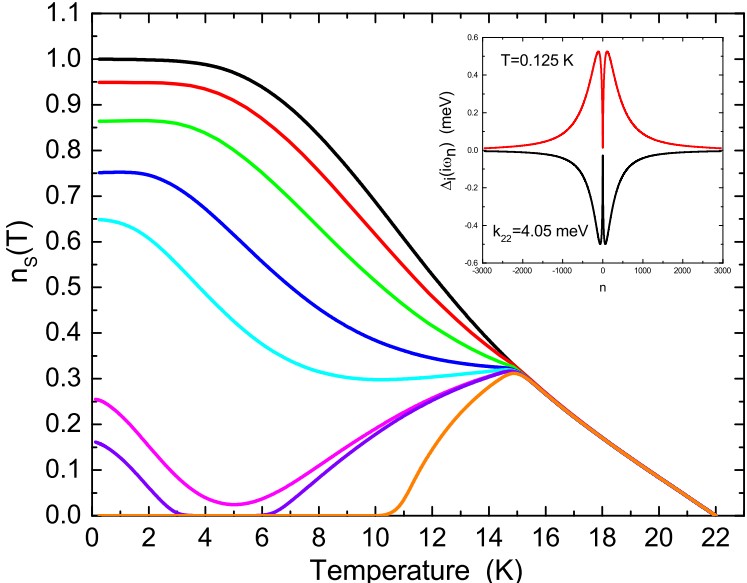

**Figure 5.** (Color online) The superfluid density $n_s(T)$ normalized at the value at T = 0 K in the case $k_{22} = 0$ in function of temperature obtained by solving the Eliashberg equations on an imaginary axis in the case $k_{11} = k_{12} = 0.2k_{22}$ and $\beta = 2$: black lines are for $k_{22} = 0$ meV, red lines are for $k_{22} = 1$ meV, green lines are for $k_{22} = 2$ meV, dark blue lines are for $k_{22} = 3$ meV, cyan lines are for $k_{22} = 3.5$ meV, magenta lines are for $k_{22} = 4$ meV, violet lines are for $k_{22} = 4.05$ meV and orange lines are for $k_{22} = 5$ meV. In the inset, the dependence, obtained by numerical solution of Eliashberg equations in the $k_{22} = 4.05$ meV case at $T = 0.125$ K, of the two order parameters $\Delta_j(i\omega_n)$ from the index $n$ is shown.

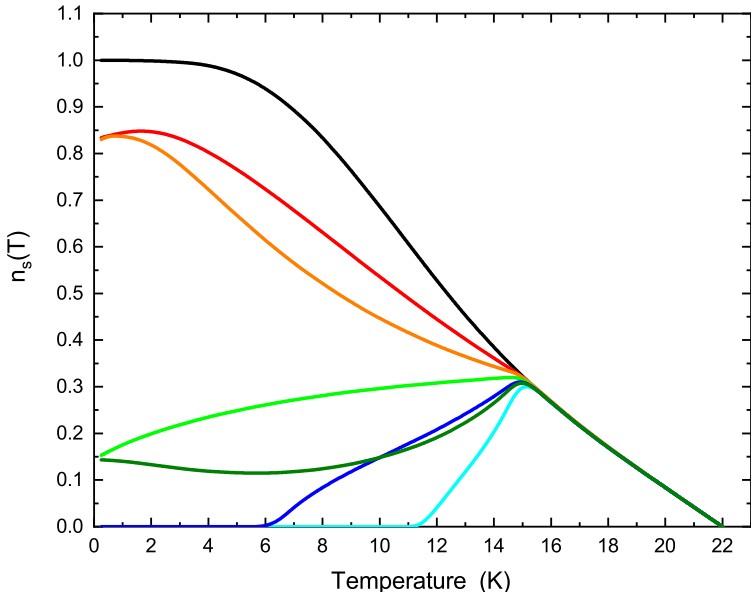

**Figure 6.** (Color online) The superfluid density $n_s(T)$ normalized at the value at T = 0 K in the case $k_{22} = 0$ in function of temperature obtained by solving the Eliashberg equations on imaginary axis in the case $k_{11} = k_{22} = 5k_{12}$: black lines are for $k_{22} = 0$ meV, red lines are for $k_{22} = 1$ meV and $\beta = 1$, orange lines are for $k_{22} = 1$ meV and $\beta = 2$, green lines are for $k_{22} = 3$ meV and $\beta = 1$, olive lines are for $k_{22} = 3$ meV and $\beta = 2$, dark blue lines are for $k_{22} = 5$ meV and $\beta = 1$, and magenta lines are for $k_{22} = 5$ meV and $\beta = 2$.

## 5. Conclusions

In conclusion, I have calculated the temperature dependence of gaps and superfluid densities for a two-band non-phononic $s\pm$-wave spin-glass superconductor. In general, the temperature dependence of superconducting properties shows a lot of different behaviours that should be observable in experiment. In this system, two competing orders modulated by temperature are present. The magnetic order breaks down superconductivity, but a complex phase diagram arises from the fact that both magnetic and superconducting coupling can depend on temperature in different ways. In addition, reentrant behavior could be a possible signature of a spin-glass state.

**Funding:** This research received no external funding

**Acknowledgments:** The author acknowledges support from the MEPhI Academic Excellence Project (contract No. 02.a03.21.0005).

**Conflicts of Interest:** The authors declare no conflict of interest.

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
