# Peer review of "Eliashberg Theory of a Multiband Non-Phononic Spin Glass Superconductor"

_magnetochemistry, doi:10.3390/magnetochemistry6040051_

Round 1

Reviewer 1 Report

This paper calculate the energy gap and "superfluid density  ns" based on the Eliashberg theory. It is known that when the gap is formed by electron pairing due to phonon mediated attraction it provides a reasonable experimental quantities of the superconductor; however it dose not work for the spin-fluctuation mediated pairing. Actually, it has been pointed out that the reason for the occurrence of superconductivity should not be considered due to the electron-pairing (see, H. Koizumi,  J.Supercond.Nov.Magn. 33, 1697, (2020)), but the appearance of the non-trivial Berry connection. This Berry connection is shown to be stabilized by the phonon mediated attraction, thus, for some superconductors, the superconducting transition temperature coincides with the energy gap formation temperature as the BCS theory predicts. 

Iron based superconductors does not obey the BCS theory; thus, it is very likely to cause erroneous predictions. At least, the author should provide the comparison with experiments. 

The theory of superconductor now needs a serious rebuilding. And the elucidation of the Iron based superconductors seems to require such a revision. The author may find something on this issue in H. Koizumi, Symmetry 12, 776, (2020); H. Koizumi, EPL 131, 37001 (2020); http://arxiv.org/abs/2008.07814.

That's all.

Author Response

This paper calculate the energy gap and "superfluid density ns" based on the Eliashberg theory.
It is known that when the gap is formed by electron pairing due to phonon mediated attraction
it provides a reasonable experimental quantities of the superconductor;
however it dose not work for the spin-fluctuation mediated pairing.
Actually, it has been pointed out that the reason for the occurrence of superconductivity
should not be considered due to the electron-pairing (see, H. Koizumi, J.Supercond.Nov.Magn. 33, 1697, (2020)),
but the appearance of the non-trivial Berry connection. This Berry connection is shown to be stabilized
by the phonon mediated attraction, thus, for some superconductors,
the superconducting transition temperature coincides with the energy gap formation temperature as the BCS theory predicts.

Iron based superconductors does not obey the BCS theory; thus, it is very likely to cause erroneous predictions.
At least, the author should provide the comparison with experiments.

ANSWER:

Yes, iron based superconductors does not obey the BCS theory but to interband Eliashberg theory that, also in the limit of weak coupling,
is different from BCS theory (Oleg V. Dolgov, Igor I. Mazin and David Parker,
Alexander A. Golubov, PHYSICAL REVIEW B79, 060502R(2009))
Yes, I have done the comparison with experiments for the iron pnictides in many papers, see for example:
G.A. Ummarino, M. Tortello, D. Daghero, R.S. Gonnelli, J. Supercond. Nov. Magn. 24, 247,
(2011); G.A. Ummarino, Phys. Rev. B 83, 092508 (2011); GA Ummarino, Sara Galasso and A Sanna, J. Phys.: Condens. Matter 25 (2013) 205701.
G.A. Ummarino, Physica C 529 50 (2016); G.A. Ummarino, A.V. Muratov, L.S. Kadyrov, B.P. Gorshunov, S. Richter, A. Anna
Thomas, R. Huhne and Y.A. Aleshchenko, Supercond. Sci. Technol. 33 (2020) 075005.
I have added these references in the paper.
For this paper I have not experimental data to compare with the theory.

QUESTION:

The theory of superconductor now needs a serious rebuilding. And the elucidation of the Iron based superconductors seems to require such a revision.
The author may find something on this issue in H. Koizumi, Symmetry 12, 776, (2020); H. Koizumi, EPL 131, 37001 (2020); http://arxiv.org/abs/2008.0781

ANSWER:

I add these references and a phrase in the way that the reader is aware that other approaches are possible to explain superconductivity in iron pnictides.
"For completeness it is necessary to mention an alternative approach to explain thesuperconductivity in the iron pnictides based on the appearance of the non-trivial Berry connection [25]."

I thank the referee for introducing me to this new theory that I did not know.

Reviewer 2 Report

In the manuscript entitled “Eliashberg theory of a multiband non-phononic spin glass superconductor” by Ummarino, the author studies the effect of spin-glass formation on the superconducting state of a two-band superconductor using Eliashberg theory. The model parameters chosen are relevant to the Eu-iron based superconductors where superconductivity has been found to co-exist with a spin-glass phase.

The manuscript covers an interesting aspect of iron based superconductors. The presentation of the model is clear, the approximations and the choice of parameters are well discussed and referenced. The results on the re-entrant superconductivity are interesting and the superfluid density predictions may well serve to motivate future experiments.

I recommend that the paper is accepted for publication. I leave it to the author to consider the few suggestions given below that I think could help improve the manuscript further.

- In line 117 the author mentions that the true physical situation for the pnictides involves three bands. Isn’t the actual iron pnictide Fermi surface made of at least four bands? Perhaps the author means the 1-Fe unit cell; in this case I think it is best if this is specified in the text.

- line 145: The author discusses the choice of impurity scattering strengths in the here assumed model where the intraband hole contribution is taken as dominant. How general is the applicability of this assumption in other pnictides? For example, inclusion of the orbital degrees of freedom has been shown to lead to an important contribution from interband impurities [see e.g. PRL 103, 177001 (2009)].

- How do the single-band results of Naas et al. and Carbotte et al. compare with the present two-band calculations (given that the modeling of the spin-glass state is similar) ? Is re-entrant superconductivity peculiar to multiband systems?

Author Response

In the manuscript entitled “Eliashberg theory of a multiband non-phononic spin glass superconductor” by Ummarino, the author studies the effect of spin-glass formation on the superconducting state of a two-band superconductor using Eliashberg theory. The model parameters chosen are relevant to the Eu-iron based superconductors where superconductivity has been found to co-exist with a spin-glass phase.

The manuscript covers an interesting aspect of iron based superconductors. The presentation of the model is clear, the approximations and the choice of parameters are well discussed and referenced. The results on the re-entrant superconductivity are interesting and the superfluid density predictions may well serve to motivate future experiments.

I recommend that the paper is accepted for publication. I leave it to the author to consider the few suggestions given below that I think could help improve the manuscript further.

QUESTION:

- In line 117 the author mentions that the true physical situation for the pnictides involves three bands. Isn’t the actual iron pnictide Fermi surface made of at least four bands? Perhaps the author means the 1-Fe unit cell; in this case I think it is best if this is specified in the text.

ANSWER:

Yes I add this phrase:

For example in the hole doped iron compounds, within the five-orbital
model [23], the Fermi surface comprises four sheets: two hole pockets around the point
(0; 0), and two electron pockets around the points (; 0) and (0; ) [24] but, usually, the
two electron bands are very similar so it is possible to approximate the real situation
with just a electron band with a density of states at the Fermi level that is the sum of
the two contributions of the two bands. For the electron doped materials the opposite
happens and it is possible to sum the contributions of the two hole bands. In both
cases, in the end, it is possible to describe the superconductor with a 3-band model [42].

QUESTION:

- line 145: The author discusses the choice of impurity scattering strengths in the here assumed model where the intraband hole contribution is taken as dominant. How general is the applicability of this assumption in other pnictides?

ANSWER:

I add another reference in the way to explain that this is a common situation in the pnictides.
Yu.A. Aleshchenko, A.V. Muratov, G.A. Ummarino, S. Richter, A. Anna Thomas, and R. Huhne,
J. Phys. Condens. Matter 32, (2020).

For example, inclusion of the orbital degrees of freedom has been shown to lead to an important contribution from interband impurities [see e.g. PRL 103, 177001 (2009)].

Yes, I have added this comment:

TIt is possible also to include orbital degrees of freedom
that lead, in the BCS formalism, to a fully gapped s  wave state very fragile [51]
against impurities while the experiments revealed that the suppression of Tc is weaker
than expected [52] for this model. The agreement with the experiment is found by
considering strong coupling e ects [53] and going beyond the Born approximation when
in the materials is present a large amount of impurities [54, 44, 60]. Indeed the pure
interband Eliashberg theory, also in the limit of weak coupling, is di erent from BCS
theory [55].

QUESTION:

-How do the single-band results of Naas et al. and Carbotte et al. compare with the present two-band calculations (given that the modeling of the spin-glass state is similar) ? Is re-entrant superconductivity peculiar to multiband systems?

ANSWER:

No, the re-entrant superconductivity appears also in the single band case, I have underlined this fact in the paper and I have added references.
"The re-entrant superconductivity appears also in the single band
case[17, 18, 19] but for a more realistic and complete model such as the one proposed
in this paper, the phase diagram is richer and more complex."

I thank the referee for his useful suggestions.

Round 2

Reviewer 1 Report

At present, this paper may be acceptable.

However, the author should worry about the upcoming reformulation that will take place in superconductivity theory.